# Mapping and spatial distribution of relict charcoal hearths across Poland

Michał Słowiński[1*], Agnieszka Halaś[1], Michał A. Niedzielski[2], Krzysztof Szewczyk[1], Jerzy Jonczak[3], Dominika Łucrów[1], Sebastian Tyszkowski[4], Sandra Słowińska[5], Agnieszka Gruszczyńska[1], Bogusława Kruczkowska[3], Aleksandra Chojnacka[6], Tomasz Polkowski[1], Krzysztof Sztabkowski[7], Dariusz Brykała[8], Jacek Wolski[9], Tomasz Samojlik[10], Adrian Kaszkiel[5], Barbara Gmińska-Nowak[8], Mateusz Kramkowski[4], Tomasz Związek[8]

[1] Department of Past Landscape Dynamics, Institute of Geography and Spatial Organization, Polish Academy of Sciences, Warszawa, 00-818, Poland

[2] Department of Urban and Population Studies, Institute of Geography and Spatial Organization, Polish Academy of Sciences, Warszawa, 00-818, Poland

[3] Department of Soil Science, Institute of Agriculture, Warsaw University of Life Sciences, Warszawa, 02-787, Poland

[4] Department of Environmental Resources and Geohazards, Institute of Geography and Spatial Organization, Polish Academy of Sciences, Toruń, 87-100, Poland

[5] Climate Research Department, Institute of Geography and Spatial Organization, Polish Academy of Sciences, Warszawa, 00-818, Poland

[6] Department of Biochemistry and Microbiology, Institute of Biology, University of Life Sciences, Warszawa, 02-787, Poland

[7] Laboratory of Natural Environment Chemistry, Forest Research Institute, Braci Leśnej 3, Sękocin Stary, 05-090, Poland

[8] Laboratory for Interdisciplinary Research into the Anthropocene, Institute of Geography and Spatial Organization, Polish Academy of Sciences, Toruń, 87-100, Poland

[9] Department of Geoecology, Institute of Geography and Spatial Organization, Polish Academy of Sciences, Warszawa, 00-818, Poland

[10] Mammal Research Institute, Polish Academy of Sciences, Białowieża, 17-230, Poland

*Correspondence to*: Michał Słowiński (michal.slowinski@geopan.torun.pl)

**Abstract.** This study presents the first national-scale spatial inventory of relict charcoal hearths (RCHs) in Poland, based on high-resolution LiDAR data and digital terrain analysis. Using a combination of manual interpretation, GIS-based feature extraction, and K-prototypes clustering, we identified and classified 634,815 RCHs across forested regions of the country. Each feature was georeferenced and categorized by size, morphological characteristics, slope position, and environmental context, including current and potential vegetation and soil types. Spatial analyses revealed significant regional differences in hearth density, with the highest concentrations found in western and south-central Poland, particularly in the Lower Silesian, Stobrawa, and Świętokrzyskie forests. Cluster analysis distinguished three major types of RCHs, differing in their environmental settings and spatial organization: (1) lowland pine-dominated clusters on gentle terrain, (2) isolated features on steep slopes in mixed forests, and (3) high-density hearth groups in elevated areas. Although large portions of the country appear devoid of RCHs, we argue that this reflects limitations in preservation and detection – due to long-term agricultural activity in lowlands and erosion in mountainous areas – rather than an actual absence of charcoal production. The resulting ReCHAR database offers a unique, open-access tool for interdisciplinary research on forest history, human-environment interactions, and early industrial landscapes. Its modular design supports further expansion, including links to historical settlements and industries reliant on charcoal, such as metallurgy, glassmaking, and tar or potash production.

## 1. Introduction

In the preindustrial era, forests and their resources were crucial to the economic development of European countries (Słowiński et al., 2024; Przybylski et al., 2025; Ellis, 2021). Initially, the primary demand was for wood as a construction material; however, over time, the focus shifted to other forest products such as potash, tar, pitch, and especially charcoal. These pyrolysis products found wide-ranging applications in daily life and various industries. Charcoal, due to its high energy value, served as a primary source in metal smelting and glass production. Potash was essential for fabric bleaching, producing glass and gunpowder; moreover, it served as a cleaning agent or even medicine (Ciceri et al., 2015; White, 1859). Wood tar was used as a wood preservative, adhesive, grease and medicine (birch tar) (Ebert, 2024). The techniques for producing these pyrolysis products evolved significantly over time. Early charcoal production methods were relatively inefficient, carried out in small pits measuring from one to five meters in diameter and from one to two meters in depth. These pits were primarily filled with branches and roots before being covered with earth and mulch (Groenewoudt, 2005; Klein and Bauch, 1983). This method represents one of the earliest known forms of charcoal extraction worldwide. Such features can be traced back to Roman times in Europe. For example, in modern-day Germany, remains dating to the third century AD have been discovered (Codreanu-Windauer, 2019). In Western Europe, from the late 10th to the early 11th centuries, these

pits were increasingly replaced by elevated constructions known as charcoal hearths. Charcoal hearths varied significantly in size and construction (Raab et al., 2015b; Oliveira et al., 2021; Witharana et al., 2018). They were essentially stacked piles of wood, ranging in diameter from a few meters to over 20 m (Raab et al., 2015a; Groenewoudt and Spek, 2016), covered with topsoil, turf, and forest litter from the surrounding area (Paschalis, 1973; Samojlik et al., 2013b; Rutkiewicz et al., 2019; Bielenin, 1959). These hearths were often encircled by a ditch or embankment approximately 40 cm in height or depth (Hirsch et al., 2020). The morphological characteristics of charcoal hearths – such as their size, shape, and function – often depended on the region, terrain, and period in which they were built (Hirsch et al., 2020). For example, hearths used for tar production were typically found in forests dominated by pine and included small peripheral depressions for collecting tar in special bowls (Cembrzyński, 2014). Hearths constructed on slopes were generally smaller, averaging from eight to ten meters in diameter, and were built on artificial mounds often reused in subsequent burning cycles (Hirsch et al., 2018). The main advantage of charcoal is its ability to reach high combustion temperatures – from 900 to as much as 2000°C (Smil, 2017). Additionally, it burns almost smoke free and has an energy value 50% higher than that of air-dried wood. It contains minimal moisture and is practically free of sulfur and phosphorus (Smil, 2017). Until the mid-19th century, charcoal remained the only widely available fuel in Europe capable of reaching the temperatures required to melt iron ore (Cleere et al., 1985). In the latter half of the 19th century, it was gradually replaced by more energy efficient fuels such as coal coke, hard coal, oil, and natural gas (Cembrzyński, 2014).

A single charcoal hearth could consume 48–100 m³ of wood (Marszałek and Kusiak, 2013). This resource hungry way of manufacturing charcoal and other pyrolysis products have had a direct impact on the environment (Hensel et al., 1978). The most immediate effect was deforestation. Increasing demand for products like potash, steel, and glass led to large scale forest depletion (Hensel et al., 1978). Massive deforestation occurred at the turn of the 18th and 19th centuries, in what is now Poland, transformed the landscape dramatically (Hensel et al., 1978). Forest cover in the Kingdom of Poland was around 30% at the beginning of the 19th century, but by 1909, it had decreased to 18.7%. Afterwards, deforested areas fell under the modern forestry management, with large areas reforested with pine or spruce monocultures, shifting the forest composition from deciduous to coniferous (Słowiński et al., 2019; Przybylski et al., 2025; Słowiński et al., 2024; Związek et al., 2023). At the same time, wetlands were drained, and former wastelands and forests were converted to farmland (Blackbourn, 2006; Surowiecki, 1811).

Charcoal production in hearths influenced soils both directly and indirectly. The direct impact involved thermal alteration (300–600°C) of soil beneath hearths during pyrolysis, leading to changes in mineral composition and loss of organic matter (Hirsch et al., 2018). These soils are rich in charcoal and often facilitate the reduction of

iron oxides to metallic iron. Typically classified as Technosols (WRB, 2022), they exhibit distinct physical and chemical properties resulting from pyrolysis byproducts (Hirsch et al., 2018; Schneider et al., 2022; Schneider et al., 2019; Jonczak et al., 2024). Indirect effects stemmed from deforestation in charcoal production areas, which enhanced surface runoff, erosion, and reduced soil moisture retention. This increased vulnerability to degradation and desertification (Pierik et al., 2018; Groenewoudt & Spek, 2016; Jonczak et al., 2024). On slopes, deforestation accelerated denudation and aeolian activity, hindering forest regeneration and plant growth (Groenewoudt & Spek, 2016). A further indirect and potentially long-lasting effect is visible in vegetation patterns. Elevated nitrogen levels under hearths, likely due to charcoal uptake, can influence plant community composition (Hirsch et al., 2017; Hardy et al., 2017). Additionally, trees growing in such soils often exhibit reduced vigour and lower timber quality (Buras et al., 2020; Hirsch et al., 2018; Buras et al., 2015).

Until recently, charcoal hearth remnants were usually encountered by chance during forestry or soil science work (Ludemann, 2012), or, in the best case, during field surveys (Samojlik et al., 2013a; Samojlik et al., 2013b). These structures appear as circular mounds or groups of mounds, often several meters in diameter across, consisting of soil mixed with charcoal particles or charcoal dust, covered with organic debris and recent sediments. Their detection has been greatly facilitated by Airborne Laser Scanning (ALS) and LiDAR (Light Detection and Ranging) technology and GIS based analysis (Suh et al., 2021; Suh et al., 2023). Such studies have been conducted in Germany (Raab et al., 2015a; Hirsch et al., 2017), USA (Suh et al., 2023; Der Vaart et al., 2022; Bonhage et al., 2023), Sweden (Davis and Lundin, 2021), France (Dupin et al., 2017), Italy (Garbarino et al., 2022), and Poland (Samojlik et al., 2013b; Rutkiewicz et al., 2019; Jonczak et al., 2024; Słowiński et al., 2022). Importantly, LiDAR allowed researchers to identify features beneath forest canopies and in areas previously inaccessible for field surveys. Previous studies have shown that hearths were rather preserved in less fertile soils, unsuitable for agriculture (Mikulski, 1994), than in areas with active cultivation. Importantly, this pattern reflects not only differential preservation under modern land use, but also the original placement of many RCHs in marginal landscapes with less productive soils, where forest use was favored over crop production. Due to that, many hearths were most likely destroyed by continuous farming practices like deep ploughing. Evidence for hearths in agricultural lands exists mainly in specific areas, such as in the Netherlands (Hardy et al., 2017).

**Turning point in the research**

Up to this point, research on charcoal hearths in Poland has primarily focused on individual sites or small groups of sites (Waga et al., 2022; Kałagate et al., 2012; Lasota et al., 2021; Samojlik et al., 2013b; Rutkiewicz et al., 2021). As a result, relatively little is known about their total number and spatial distribution on a much larger scale (e.g. the whole country), or the environmental conditions in which they occur. Given that the number and density

of such features can have important environmental implications, it is essential to maintain an open-access database containing their locations and basic attributes. The main objective of this study is therefore to develop a comprehensive database detailing the location, size, density, and environmental context of relict charcoal hearths (RCHs) in Poland (Słowiński et al., 2025). In addition, the research aims to identify regional differences in charcoal production practices, with a focus on morphological characteristics, site location, and key environmental variables such as slope and proximity to water. Although the number of identified charcoal hearths offers great scientific potential, meaningful analysis would not be possible without first constructing a basic database. Such a database serves as a fundamental resource for broader regional considerations such as the historical functioning of forest ecosystems as well as more local topics, such as site specific degradation and disturbance processes. To achieve this, we analyzed a region in northern Poland covering 186,905.7 km², which includes 60,107.9 km² of forested land. This effort let to the creation of the first nationwide database documenting RCH in Poland. The mapped sites were subsequently analyzed using geostatistical methods, linking the density and spatial distribution of RCHs to environmental conditions and potential vegetation. This newly developed database facilitates the investigation of regional variability in charcoal production techniques. Until now, it remained unclear whether distinct types of charcoal hearths existed, and to what extent production methods varied either regionally or as a function of hearth morphology. To address this knowledge gap, we applied clustering techniques to classify charcoal hearths based on both morphological and environmental parameters.

## 2. Data and methods

### 2.1. Study site

This study covers forested areas across the entire territory of Poland (Fig. 1). Outside forests particularly in farmland and other intensively transformed landscapes the detectability of charcoal hearth microrelief in ALS-derived Digital Terrain Model (DTMs) is very low and in many cases not feasible. Agricultural operations (ploughing, levelling, drainage, earthworks) and subsequent construction repeatedly disturb or obliterate these small relief features; therefore, our mapping was restricted to contemporary forests where such features are most likely to be preserved. As of 2023, forests in Poland covered approximately 9.3 million hectares, which corresponds to 29.6% of the country's land area (Forest Data Bank: *https://www.bdl.lasy.gov.pl/portal/"*). Forest cover has increased significantly over the past century, rising from 21% in 1945 to nearly 30% today (Broda, 2000). This expansion has been largely driven by afforestation initiatives on former agricultural lands, particularly in lowland and central regions. The vast majority of forests in Poland are publicly owned. In 2021, public forests accounted for 80.7% of all forested land, with 76.9% managed by the State Forests National "Forest Holding" (Pol. *Państwowe Gospodarstwo Leśne "Lasy Państwowe"*). Forest composition is dominated by Scots pine (*Pinus*

*sylvestris*), which represents approximately 58.5% of all stands. These pine dominated monocultures, often established on poor sandy soils or reclaimed farmland, define much of the lowland forest landscape. Other tree species include birch (*Betula* spp.), oak (*Quercus* spp.), and beech (*Fagus sylvatica*), with spruce (*Picea abies*) and fir (*Abies alba*) commonly found in the mountainous regions (Matuszkiewicz, 1999).

Poland's geomorphology is the result of a complex history of glacial and tectonic processes. The northern and central lowlands were formed through a succession of Pleistocene glaciations particularly the Vistulian (Weichselian), Wartan, and older Odra and Sanian phases (Marks, 2005; Marks, 2002). These events left behind a wide range of glacial landforms including moraines, eskers, sandurs (outwash plains), and kettle lakes. In contrast, the southern part of the country is defined by tectonic uplift and erosion processes associated with the Sudetes and Carpathian Mountain ranges, leading to a more rugged and elevated landscape (Solon et al., 2018). The diversity of soils across Poland reflects this geomorphological variability. In the lowlands, particularly in the north and center, sandy and acidic soils are predominant. These include Arenosols (mainly Brunic Arenosols) and Podzols— soils typical for coniferous stands on fluvioglacial, glacial and aeolian sandy substrates (e.g. Jankowski et al., 2011; Kabała et al., 2021). In upland and mountainous regions, Cambisols (brown earths) are widespread, developed from loess, weathered rocks, or glacial tills, often supporting mixed or deciduous forests (Gałka et al., 2013; Kacprzak and Derkowski, 2007). Rendzinas, shallow calcareous soils developed from limestone and marl, are found in smaller patches, particularly in southern uplands, and support unique plant communities (Lasota et al., 2018; Miechówka and Drewnik, 2018). Climatically, Poland lies within the humid continental zone (Dfb according to the Köppen classification), characterized by cold winters and warm summers. The average annual temperature ranges from 7°C to 9°C (Kottek et al., 2006). Annual precipitation is regionally differentiated, ranging from about 500 mm in the central lowlands to over 1,200 mm in mountainous regions (Institute of Meteorology and Water Management - National Research Institute). This combination of historical land use, ownership structure, glacial and tectonic legacy, and environmental heterogeneity provides a complex yet coherent context for analyzing the distribution of relict charcoal hearths (RCHs) across Polish forest landscapes.

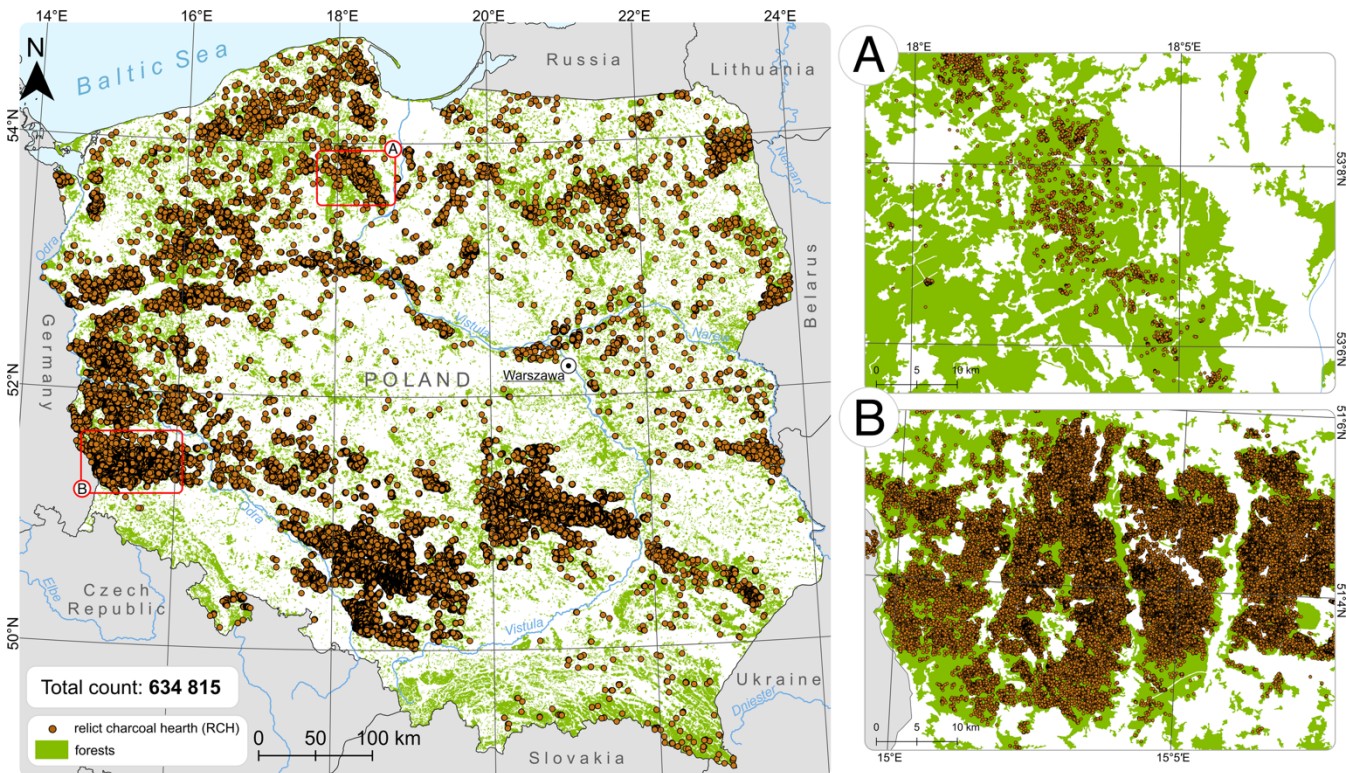

**Figure 1.** Location of the study area within Poland with all identified relict charcoal hearths (RCHs). The figure includes zoomed-in illustrations showing forest cover based on CORINE Land Cover 2018 (European Environmental Agency). Panels (A) and (B) present contrasting densities of RCH distribution across forested regions of Poland.

### 2.2. Data collection

To identify and map all relict charcoal hearths (RCHs), we processed a DTM derived from light detection and ranging (LiDAR) data in QGIS 3.18. The DTM was inspected using relief-shaded visualizations (single- and multi-azimuth hillshade and slope shading) generated directly in QGIS. For a consistent and reproducible cartographic backdrop, we also consumed a public web map tile service (WMTS) provided by the national geoportal (https://www.geoportal.gov.pl) under its full name "The viewing of shaded relief for a digital terrain model" (Geopartal, 2025).

A transparent grid with a resolution of 500 × 500 meters was superimposed across the entire forested area (Fig. 2). This ensured comprehensive spatial coverage and minimized the risk of omissions during the manual interpretation of features. Charcoal hearths were visually recognized by their circular morphology and often occurred in dense clusters sometimes comprising several hundred objects within a single grid square.

Two primary morphological types were identified. The first type, commonly located on flat terrain, typically exceeded 15 meters in diameter and was often surrounded by a visible ditch. The second type was constructed on specially prepared platforms on slopes; these hearths were generally smaller (under 10 meters in diameter), lacked surrounding ditches, and appeared less prominently on LiDAR-derived surfaces. For analytical purposes, charcoal hearths were classified according to: (1) Size — small (<10 m), medium (10–15 m), and large (>15 m); (2) Surface context and function — based on both topography and evidence of extracted material. Three categories were distinguished: (A) Hearths located on flat terrain; (B) Hearths on flat terrain associated with tar extraction, often identifiable by peripheral ditches several centimeters wide and deep; (C) Hearths located on slopes (Fig. 3). Charcoal hearths were exclusively identified within forested areas (Słowiński et al., 2025). In agricultural landscapes, these features have likely been destroyed due to soil disturbance caused by deep ploughing and other farming practices (Hardy et al., 2017). To verify the reliability of the identification process, a random sample of 50 grid squares was reviewed manually.

Each relict charcoal hearth (RCH) was assigned a set of attributes to support further analysis. These included, in addition to the previously mentioned size, current land cover based on CORINE Land Cover 2018, level 3 (European Environment Agency, 2020), potential natural vegetation (Matuszkiewicz and Wolski, 2023), altitude and slope derived from the Digital Elevation Model (DEM), (European Space Agency, 2000), as well as distances to the nearest other RCH and the nearest water body. Only rivers and lakes were considered in the latter, based on the Hydrographic Map of Poland (MPHP). Soil type information was obtained from two sources: the Forest Data Bank, following the Polish Soil Classification (Kabała et al., 2019), and the European Soil Database, based on the World Reference Base for Soil Resources (WRB) and FAO85 classification systems (Panagos, 2006; Liedekerke et al., 2006). Additionally, information on contemporary forest ecosystems was also derived from the Forest Data Bank.

To assign environmental characteristics to each relict charcoal hearth (RCH), spatial joins were performed in QGIS 3.16.2 using the Join attributes by location (summary) tool. The point layer representing RCHs was used as the target layer, and relevant polygon layers (e.g., land cover, soil type, and potential natural vegetation) served as join layers. The spatial relationship intersects was applied to associate each point with the polygon within which it was located. In cases where a point fell within multiple overlapping polygons, the first matching record was selected. If a point did not fall within the extent of a given polygon layer, a placeholder value of 9999 was assigned to the corresponding attribute. The resulting output layer included the original RCH attributes along with the relevant information extracted from the polygon layers.

225 To calculate the distance to the nearest neighboring RCH, the Distance to nearest hub (points) tool was used. The RCH layer was applied as both the source and destination layer, with a unique identifier field specified to distinguish individual points. The option to exclude each source point from its own candidate list was enabled to avoid self-matching. The output provided the Euclidean distance from each RCH to its nearest neighboring RCH, measured in meters.

230 Additionally, to determine the distance from each RCH to the nearest water body – including both rivers (line features) and lakes (polygon features) – the Distance to nearest hub (points) tool was again applied. The RCH layer was set as the source, and the vector version of the MPHP served as the destination layer. The tool calculated the shortest Euclidean distance from each point to the closest hydrographic feature, measured in meters.

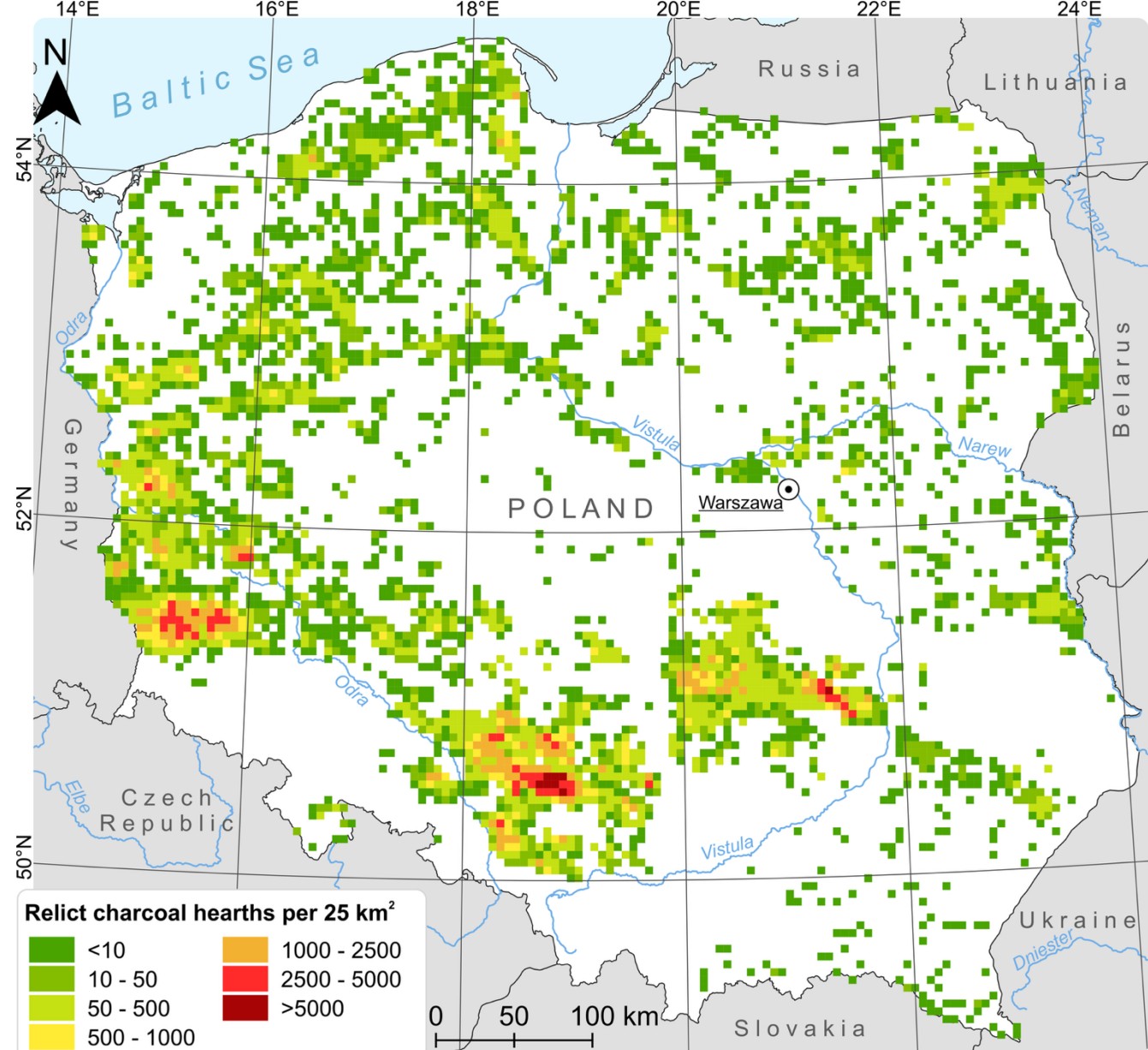

**Figure 2.** Map of the spatial density of relict charcoal hearths across forested areas of Poland

### 2.3. Clustering

There are two main categories of clustering techniques: hierarchical and partitioning. The hierarchical approach creates clusters by iteratively grouping clusters with similar values, using a distance tolerance. In this method, each observation initially forms its cluster, and the number of final clusters is not predetermined. On the other hand, the

partitioning approach iteratively reassigns observations to clusters based on a distance tolerance, starting with initial, randomly placed cluster centroids. In this method, the number of clusters is predefined. Typically, the partitioning approach is tested across a range of cluster values, and several statistics are used to determine the optimal number of clusters for further analysis. Choosing between these techniques is largely a matter of choice preference; however, the partitioning approach is generally less computationally demanding and more efficient for handling large data sets (Rokach and Maimon, 2005). In both cases, a decision regarding the number of clusters is required—either when to stop in the hierarchical case or what initial number to set in the partitioning case. Given the large number of observations under consideration, as discussed below, the partitioning approach has been chosen to define charcoal hearth clusters. The selection of a specific partitioning algorithm is discussed in a subsequent section.

Clustering is carried out using six categorical (hearth diameter, land cover, potential vegetation, soil type, forest site type, tree species) and four numerical (distance to other charcoal hearths, distance to water, altitude, slope) morphological and environmental variables. Importantly, geographical coordinates are not used as input in the clustering analysis. To account for different measurement scales, the numerical variables are standardized using z-scores. Given the heterogeneous data types, the K-prototypes partitioning algorithm is employed (Huang, 1998; Rokach and Maimon, 2005). This algorithm combines the features of the K-means algorithm, which is designed for numerical variables, and the K-modes algorithm, which is geared toward categorical variables. The K-prototypes approach calculates two separate dissimilarity measures: one aims to minimize the sum of squared Euclidean distances between numerical values and their corresponding cluster means, while the other seeks to minimize the number of mismatches between categorical values and their respective cluster modes. Each mismatch increments the distance score by one. The overall dissimilarity measure is then obtained as a weighted sum of these K-means and K-modes dissimilarity scores. Clustering was conducted over a range of clusters, specifically from 2 to 8, using the k-modes package in Python.

## 3. Results

### 3.1. Characteristics of charcoal hearths

In the analyzed area, a total of 634,815 relict charcoal hearths (RCHs) were identified and mapped (Fig. 2). However, these features are preserved only in certain forested regions, typically those with limited anthropogenic disturbance. In other areas, traces of similar hearths have likely been lost due to erosion caused by agricultural activity, construction, or modern forestry practices. Three distinct types of charcoal hearths were identified: Type A, Type B, and Type C (Fig. 3, S1).

Types A and B are morphologically similar. Both present as convex, circular mounds rising up to 30 cm above ground level, with diameters ranging mostly between 5 and 20 meters. Each form is typically encircled by a shallow ditch, reaching depths of up to 40 cm. Charcoal fragments-sometimes exceeding 10 cm in size-are generally found a few centimetres below the surface. The key distinction between Type A and Type B hearths lies in the presence of peripheral depressions.

Type B hearths exhibit 3 to 6 depressions, each several tens of centimetres in diameter, located near the surrounding ditch. These depressions were used for tar collection; ceramic or metal bowls (Fokt, 2012) were placed in them to gather tar, which would flow out during the pyrolysis of resin-rich wood, typically coniferous species such as *Pinus* spp.

Type C hearths differ in both size and location. They are smaller (typically 8–12 meters in diameter) and were built on specially constructed platforms on slopes to ensure a level combustion surface. In the study area, Type C hearths were recorded only in the northern region—in the Kartuski Forest—on moraine hills overlooking the Bay of Gdańsk (west of Gdańsk and Puck), within predominantly beech (*Fagus sylvatica*) forests. Uniquely, these are the only hearths documented in the field where hardwoods, rather than pines, were used. Due to slope erosion, charcoal remnants from these hearths often lie directly on or just beneath the soil surface.

## 3.2. Mapping and spatial distribution of charcoal hearths

### 3.2.1. Size of charcoal hearths and their location (slope)

Across the study area, researchers identified and examined 634,815 relict charcoal hearths. The majority of RCHs were classified as medium-sized (10–15 m in diameter), representing 54.2% of the total. Small hearths (<10 m) accounted for 35.3%, while large hearths (>15 m) made up 10.5%. Most hearths were located on gently sloping terrain, with frequency decreasing as slope increases. Specifically, 12.4% of hearths were found on slopes <1°, 43.9% on 1°, 23.4% on 2°, 10.8% on 3°, 4.9% on 4°, 2.3% on 5°, 1.1% on 6°, 0.5% on 7°, and 0.3% on 8°. Only 0.4% occurred on terrain with slopes between 8° and 58°, and just 46 hearths (0.007%) were located on slopes exceeding 20°.

### 3.2.2. Charcoal hearths and their location in relation to potential and actual vegetation

Charcoal hearths were predominantly located in contemporary forest ecosystems dominated by Scots pine, which accounts for 84.8% of the forest stands where these features occur. Other common tree species include birch (*Betula spp.*, 3.2%), oak (*Quercus spp.*, 1.9%), and beech (*Fagus sylvatica*, 1.5%). Spruce, larch, and fir are less common, each representing 0.4% of the total. Based on current land cover, 80.3% of charcoal hearths are found in coniferous forests, 9.7% in mixed forests, 5.6% in transitional woodland-shrub vegetation, 3.6% in broad-leaved forests, 0.4% in natural grasslands, approximately 0.1% in pastures, and remaining 0,2% in other land cover classes. Overlaying

hearth locations with the map of potential natural vegetation (Matuszkiewicz et al., 1995; Matuszkiewicz, 2008; Matuszkiewicz and Wolski, 2023) revealed considerable variation in historical forest types. The majority of hearths—41.2% (261,550)—occur in areas originally classified as suboceanic Middle-European pine forest. Other

305   significant types include continental oak-pine mixed forest (26.4%; 167,465 hearths), acidophilous oak forest (11.9%; 75,310), and subcontinental colline lime-oak-hornbeam forest (4.1%; 26,329). Additional forest types are represented in smaller proportions, including lowland alder and ash-alder forest (3.5%), Sarmatian oak and pine-oak forest (3.5%), acidophilous beech-oak forest (1.5%), and lowland acidophilous beech forest (1.2%). Other units include submontane fir forests, alder fen forests, forb-rich beech forests, swamp pine forests, and various regional

310   oak-hornbeam communities. In 0.2% (1,654) of the mapped hearths, the potential vegetation type could not be determined due to incomplete or unavailable data.

### 3.2.3. Charcoal hearths and soil types

Charcoal hearths were found on nine distinct soil types. The largest shares were located on Brunic Arenosols (Protospodic), accounting for 29.7% of all features, followed by Albic Podzols at 22.9%. Other soil classes included

315   Brunic Arenosols (12.6%), Gleyic Albic Podzols (8.6%), Brunic Arenosols II (5.8%), Podzolic soils (2.4%), Umbric Podzols (1.7%), Gleysols (1.5%), and Dystric Cambisols (1.2%). In 6.2% of the cases, the soil type could not be determined due to insufficient or missing data.

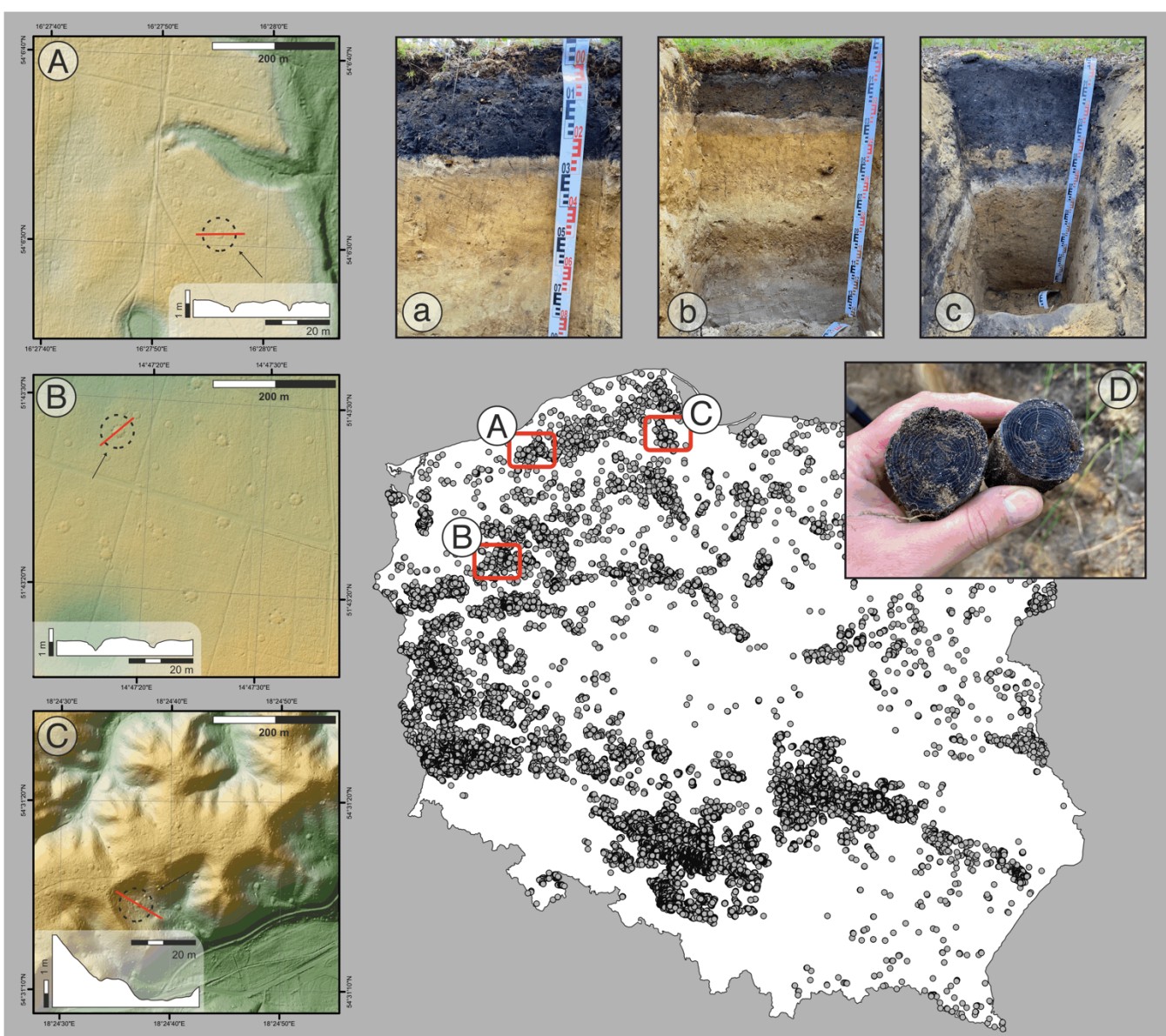

**Figure 3.** Map of Poland with all identified Relict Charcoal Hearths with a schematic cross-section of three typical hearth configurations: (A) hearth located on flat terrain with visible hollows used for tar collection, (B) hearth built on a gentle slope without tar-extraction features, and (C) hearth situated on a steep incline, reflecting adaptation to challenging topographic conditions. Soil profiles (a), (b), and (c) correspond to field-recorded sections for each hearth type. Picture (D) shows macroscopic charcoal fragments preserved in the soil profile.

### 3.2.4. Density of charcoal hearths

The spatial distribution of charcoal hearths across the study area shows considerable variation in density (Fig. 2, S2 ). Local densities range from as low as 1 to as high as 9,230 hearths per 25 km², with an average of approximately 181 hearths per 25 km². In northern and northeastern Poland, densities generally remain below 50 hearths per 25 km², with only isolated hotspots reaching between 500 and 2,500 hearths per 25 km². In contrast, higher densities were observed in the western and southwestern regions of the study area. The highest concentrations of charcoal hearths were recorded in the Lower Silesian Forests (Bory Dolnośląskie), Stobrawa Forests (Bory Stobrawskie), and the Świętokrzyskie Forest (Puszcza Świętokrzyska). In these areas, local densities often exceeded 2,500 hearths per 25 km² and, in some locations surpassed 5,000. Additionally, we mapped and reported charcoal-hearth density using a finer $1 \times 1$ km grid, hearths per km² (S1).

### 3.2.5. Validation and limitations in the detectability of charcoal hearths

The spatial distribution of RCHs across Poland reveals several regions with no detected features, forming so-called "blank zones" in the dataset. Crucially, such absences should not be interpreted as evidence of no historical charcoal production, but as a consequence of preservation state and the detection limits of bare-earth, LiDAR-derived DTMs. In many places the characteristic rim-and-plateau microrelief has been degraded below our operational relief threshold (~0.15 m), precluding reliable detection particularly in long-term agricultural lowlands, where repeated ploughing, levelling, drainage and soil homogenization obliterate or mask these small forms, and in steep mountain settings, where erosion, colluviation and other mass wasting processes displace or bury remains beyond the reach of surface based remote sensing. To minimize false positives, we required a conservative morphological signature: round shape, flat top with a subtle peripheral rim, diameter 6–25 m and detectable local-relief amplitude in the bare-earth DTM. Features that did not meet the mandatory shape + flat-top criteria, or modern earthworks, were flagged as ambiguous and excluded from the analysis. This interpretation is supported by archival and historical sources documenting charcoal-related activity in regions where no RCHs are currently visible in the DTM data. Notable examples include the Mazovia region, the Bieszczady Mountains, the Karkonosze, the Izera Mountains, and other parts of the Sudetes, where charcoal production was historically linked to glassworks, iron smelting, tar distillation, and potash manufacture all industries reliant on large quantities of wood and charcoal (Słowiński et al., 2024). Although systematic country-wide field validation is infeasible, we conducted targeted checks during multiple field campaigns across different regions [n=9]: at accessible mounds we verified the flat-topped geometry and exposed charcoal-rich layers in shallow test pits and hand auger cores. To date, all [n=7] visited DTM-identified features [n>250] have been confirmed as RCHs in the field."

### 3.2.6. Cluster analysis

After running the k-prototypes algorithm for 2 to 8 clusters, diagnostic tools suggested that a three-cluster solution would be most appropriate. Although the scree plot—representing the sum of the combined distances from all points to their respective cluster centroids (cost)—for varying numbers of clusters showed a close to linear decline in cost and thus did not show an obvious "elbow" where additional clusters barely change the cost which would indicate the optimal number of clusters (Fig. 4). The largest elbow occurs at a five cluster solution, with the second largest elbow at a three cluster solution. However, both the silhouette statistics and visual inspections of maps indicated that a three-cluster solution would be optimal. The silhouette coefficient compares the mean intracluster and the mean nearest-cluster distance. Higher values indicate denser and better-separated clusters. Silhouette values (Fig. 4) for two and three clusters were nearly identical; however, a three-cluster solution was preferred because one of the clusters in the three-cluster solution, with only 338 hearths, appears in all cluster solutions except two clusters, which clearly is distinctive and important. Furthermore, a three cluster solution has a positive value indicating better clustering while a five cluster solution has a negative value indicating poor separation of clusters. This three-cluster preference is further corroborated by a visual inspection of cluster maps (a map of a three-cluster only solution is presented below for brevity). Starting from a four-cluster solution, each new cluster merely becomes a spatial subset of one of the two large clusters evident in the three-cluster solution. For example, the third cluster in a three-cluster solution is divided into two clusters in a four-cluster solution, and the first cluster in the three and four-cluster solutions is divided into two clusters in the five-cluster solution. In other words, spatial intermixing of clusters never occurs as the number of clusters increases.

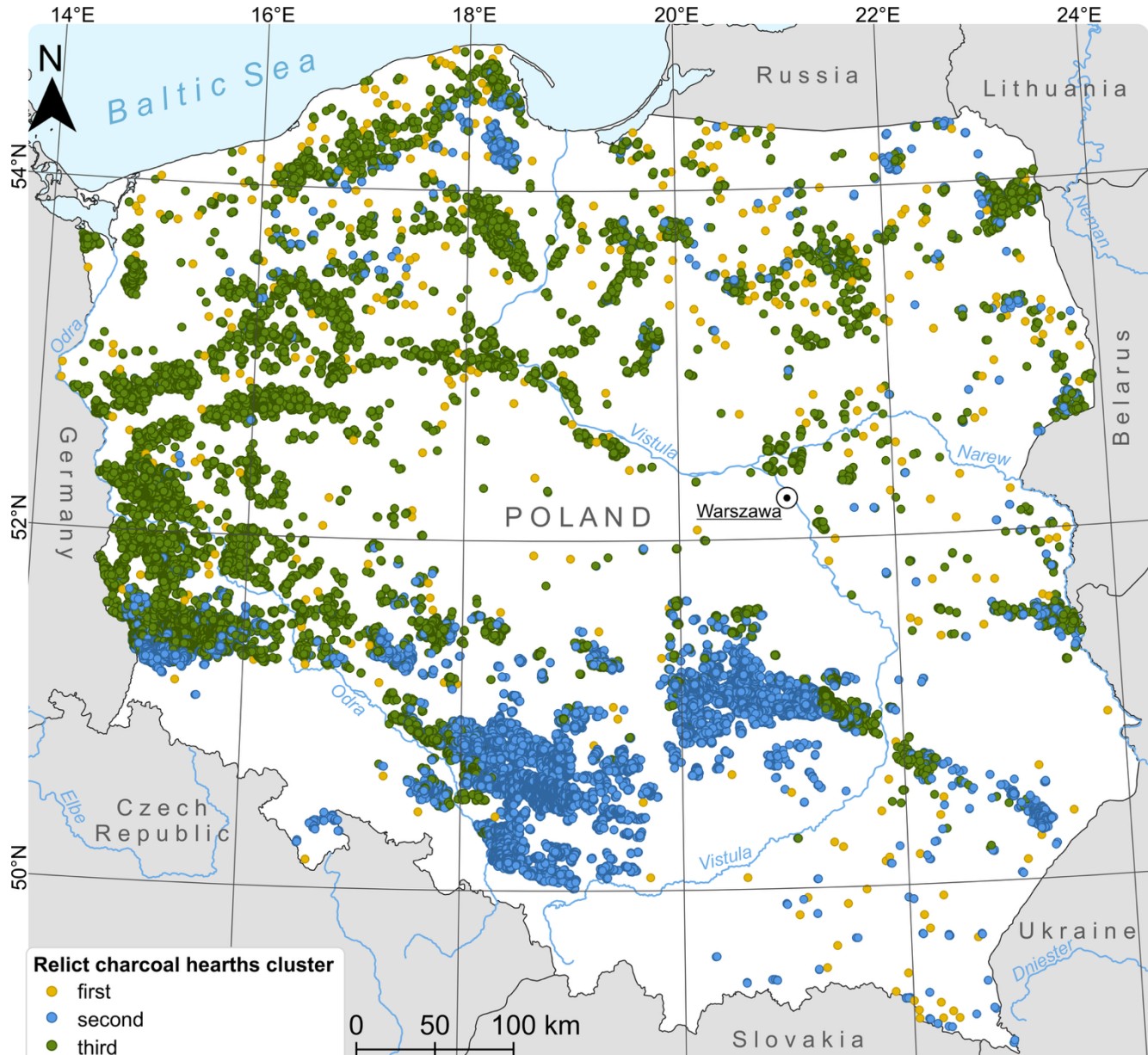

**Figure 4.** Spatial distribution of three identified types of charcoal hearth clusters across Poland

Additional support for a three-cluster solution comes from an analysis of cluster characteristics. Table 1 provides details on these characteristics, based on mode values and mean z-scores. Cluster 2 is distinct from the others, as it contains only 338 hearths and differs in potential vegetation. Moreover, the characteristics of the clusters beyond the three-cluster solution are driven mainly by numerical variables, as the mode values remain largely unchanged. In the three-cluster solution, clusters 1 and 3 share similar environmental characteristics, and these characteristics

remain largely constant regardless of the number of clusters considered, further validating the suitability of a three-cluster approach.

Cluster characteristics are based on modes as well as the highest and lowest z-scores. Cluster 1 contains medium-sized hearths located in pine forests, close to other hearths, far from rivers and lakes, on gentle slopes and at low altitudes. Cluster 2 contains medium-sized hearths found in mixed forests, isolated from other hearths, and on steep slopes. Cluster 3 contains medium-sized hearths located in pine forests, in close proximity to other hearths, and at high altitudes on steeper slopes.

**Table 1.** Modes and z-score means (minimum and maximum values highlighted) across clusters for a three-cluster solution.

| Variables | Cluster 1 | Cluster 2 | Cluster 3 |
|---|---|---|---|
| Cluster size | 338 | 324 903 | 264 694 |
| Hearth diameter | 10–15 m | 10–15 m | 10–15 m |
| Corine Land Cover | Coniferous forests | Coniferous forests | Coniferous forests |
| Potential vegetation | Continental mixed pine-oak forests | Continental mixed pine-oak forests | Suboceanic pine forest |
| Dominant Soil Reference Group (WRB) | Brunic Arenosol | Brunic Arenosol | Brunic Arenosol |
| Forest site types | Fresh mixed forest | Fresh mixed coniferous forest | Fresh coniferous forest |
| Tree species | Scots pine | Scots pine | Scots pine |
| Distance to hearth | 31.777 (5 744 m) | −0.058 (41 m) | 0.029 (56 m) |
| Distance to water | -0.131 (1 055 m) | -0.429 (752 m) | 0.527 (1 726 m) |
| Altitude | -0.563 (158 m) | 0.706 (249 m) | -0.865 (136 m) |
| Slope | 0.610 (2.60) | 0.179 (1.97) | -0.219 (1.39) |

## 4. Data availability

All data generated and analyzed during this study are publicly available through the ZENODO repository: https://doi.org/10.5281/zenodo.15630690 (Słowiński et al., 2025). The dataset includes raw and processed results, metadata, and relevant documentation to ensure reproducibility. The ReCHAR database underlying this study is hosted and maintained by the Institute of Geography and Spatial Organization, Polish Academy of Sciences. This database is openly accessible; for inquiries regarding dataset structure, access support, or additional details, please contact michal.slowinski@igipz.pan.pl and aj.halas@twarda.pan.pl.

## 5. Conclusion and future developments

The database developed in this study is the first comprehensive, high-resolution spatial dataset documenting remnants of historical charcoal production in the form of Relict Charcoal Hearths across an entire country (Słowiński et al., 2025). To our knowledge, there is currently no equivalent dataset of this scale and resolution available elsewhere in Europe or globally. While regional charcoal hearth inventories do exist, none offer the systematic national coverage, environmental context, and object-level detail provided here.

The ReCHAR dataset provides substantial value across a wide range of disciplines, including soil science, forestry, ecology, environmental history, archaeology, anthropology, and cultural heritage studies. It not only preserves evidence of past forest use but also enables new insights into long-term human impacts on forest ecosystems, soil transformation processes, and landscape change.

The current version of the database includes 634,815 georeferenced entries, categorized by hearth size (small: <10 m; medium: 10–15 m; large: >15 m), and enriched with information on current and potential vegetation, topography, proximity to water bodies, and soil types. The dataset was intentionally designed with a simple structure to ensure broad compatibility, but its modular design allows for continuous expansion and integration with other spatial datasets.

Importantly, future iterations of the ReCHAR database will also aim to incorporate other data types—such as the historical names and locations of villages, hamlets, or industrial sites directly associated with charcoal production and related industries. These include glassworks, iron smelters, potash and tar distilleries—industries that would not have existed without wood and, later, charcoal as a primary fuels and chemical inputs. By connecting spatial traces of forest exploitation with the socio-economic infrastructure it sustained, the database will support deeper investigations into early industrialization, rural economies, and environmental transformations.

The long-term vision of the ReCHAR project is to develop an open-access, community-driven research platform supporting the documentation, analysis, and exchange of data related to anthropogenic forest disturbances and spatial landscape history. Development priorities include: (1) integration of interactive visualization tools, (2) a user-friendly interface for data submission and review, and (3) the inclusion of extended metadata to enable more detailed scientific analyses.

We invite researchers, practitioners, and forest managers to contribute additional datasets, share their expertise, and co-develop analytical tools using this resource. Ultimately, ReCHAR is envisioned as a foundational platform for interdisciplinary, spatially informed research into human-environment interactions over long timescales—locally, regionally, and globally.

## 6. Author Contributions

**KSz** mapped charcoal hearths from lidar data, analyzed the data, authored or reviewed drafts of the paper, and approved the final draft.

**AH** analyzed the data, prepared figures and tables, authored or reviewed drafts of the paper, and approved the final draft.

**MN** analyzed the data and performed cluster analysis, prepared figures and tables, authored or reviewed drafts of the paper, and approved the final draft.

**DŁ** analyzed the data, authored or reviewed drafts of the paper, and approved the final draft,

**ST** analyzed the data, authored or reviewed drafts of the paper, and approved the final draft,

**AG** analyzed the data, authored or reviewed drafts of the paper, and approved the final draft,

**JJ** analyzed the data, authored or reviewed drafts of the paper, and approved the final draft,

**BK** analyzed the data, authored or reviewed drafts of the paper, and approved the final draft,

**ACh** analyzed the data, authored or reviewed drafts of the paper, and approved the final draft,

**SS** analyzed the data, authored or reviewed drafts of the paper, and approved the final draft,

**TP** analyzed the data, authored or reviewed drafts of the paper, and approved the final draft,

**KSz** analyzed the data, authored or reviewed drafts of the paper, and approved the final draft,

**DB** analyzed the data, authored or reviewed drafts of the paper, and approved the final draft,

**JW** proposed website and repository description for ReCHAR, authored or reviewed drafts of the paper, and approved the final draft,

**TS -** analyzed the data, authored or reviewed drafts of the paper, and approved the final draft,

**TZ -** analyzed the data, authored or reviewed drafts of the paper, and approved the final draft,

**BG-N** analyzed the data, authored or reviewed drafts of the paper, and approved the final draft,

**AK** analyzed the data, authored or reviewed drafts of the paper, and approved the final draft,

**MK** analyzed the data, authored or reviewed drafts of the paper, and approved the final draft,

**MS** conceived and designed the project, analyzed the data, prepared figures and tables, authored or reviewed drafts of the paper, and approved the final draft.

## 7. Acknowledgments

We would like to express our sincere gratitude to all individuals and institutions who contributed to the realization of this research. We extend our special thanks to the staff of the Darwieński National Park. for their kindness, support, and understanding during the course of the fieldwork. We also wish to thank the State Forests (Lasy Państwowe) for their assistance and for their positive attitude toward the research activities carried out. Special thanks go to Dorota Żelechowska for her commitment and support during her internship, particularly in the initial development of the research database. Finally, we would like to note that, to improve the clarity and linguistic consistency of selected parts of the text, we used a language tool based on artificial intelligence (ChatGPT). These edits were purely linguistic and did not affect the scientific content of the work.

## 8. Financial support

The research was funded by the National Science Centre projects, "Impact of charcoal production on environmental changes in Northern Poland - a novel "multi-proxy" approach" (no. 2018/31/B/ST10/02498) and "Anthropogenic transformations of the environment in the context of modernization processes of the Congress Poland" (no. 2022/47/D/HS3/02947) and "Forest of the Anthropocene: the role of Pinus sylvestris in the formation of modern states, societies and environments in the 18th and 19th centuries" (no. 2024/55/I/HS3/02551).

## 9. Figure captions

**Figure 1.** Location of the study area within Poland with all identified relict charcoal hearths (RCHs). The figure includes zoomed-in illustrations showing forest cover based on CORINE Land Cover 2018 (European Environmental Agency). Panels (A) and (B) present contrasting densities of RCH distribution across forested regions of Poland.

**Figure 2.** Map of the spatial density of relict charcoal hearths across forested areas of Poland

**Figure 3.** Map of Poland with all identified Relict Charcoal Hearths with a schematic cross-section of three typical hearth configurations: (A) hearth located on flat terrain with visible hollows used for tar collection, (B) hearth built

on a gentle slope without tar-extraction features, and (C) hearth situated on a steep incline, reflecting adaptation to challenging topographic conditions. Soil profiles (a), (b), and (c) correspond to field-recorded sections for each hearth type. Picture (D) shows macroscopic charcoal fragments preserved in the soil profile.

**Figure 4.** Spatial distribution of three identified types of charcoal hearth clusters across Poland

**Table 1.** Modes and z-score means (minimum and maximum values highlighted) across clusters for a three-cluster solution.

## 10. Supplementary Figure

**Supplementary Figure S1**. Visibility of Relict Charcoal Hearth microrelief in ALS-derived bare-earth DTM: effect of illumination azimuth (45°, 135°, 225°, 315°) and vertical exaggeration (Z = 1–4); example site (54°6′40″N; 16°28′E).

**Supplementary Figure S2**. Map of the spatial density of relict charcoal hearths across forested areas of Poland (hearths km$^{-2}$).

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
