# Peer review of "Mapping and spatial distribution of relict charcoal hearths across Poland"

_Earth System Science Data, 2025_

## Author Response (AR1)

Dear Editor,

Thank you very much for your support and for handling our manuscript throughout the review process. We greatly appreciate the time and effort invested by you and the reviewers.

We have now introduced substantial revisions in line with the reviewers' comments. In addition, we prepared a new supplementary component presenting the spatial density of relict charcoal hearths using a finer 1 × 1 km grid (hearths per km²).

As I mentioned in my previous message, during the revision my colleague Dr Mateusz Kramkowski has provided substantial assistance and in developing extensions to the manuscript, including graphical analyses. Given his significant contribution to the revised version and the final form of the paper, I would like to kindly ask whether it would be possible to add him as a co-author at this stage of the editorial process.

Sincerely,
Michał Słowiński

Thank you very much for your careful reading of our manuscript and for your positive and constructive evaluation of both the study and the released dataset. We greatly appreciate your recognition of the novelty and relevance of producing a national-scale inventory of relict charcoal hearths, as well as your insightful suggestions on how to strengthen the manuscript for an international readership. Below, we respond to each comment point-by-point and indicate the revisions we have made in the revised manuscript.

**General comment**: We are grateful for the Reviewer's encouraging assessment and for framing the broader significance of land-use legacy landforms (LULLs). We agree that the dataset can be an important step toward broader, transnational syntheses, and we hope it will also help catalyze new collaborations and coordination efforts across regions.

**Specific comments:**

1) Line 48 ff: To address the internal readership, it would be worth mentioning that this was also of great importance in the USA. Studies have especially been published in New England (e.g. https://doi.org/10.1016/j.apgeog.2023.103121). Some of these studies present state-wide inventories comparable to those presented in the paper: https://doi.org/10.1002/arp.1889, https://doi.org/10.1016/j.catena.2023.107426.

**Response**: Thank you for this excellent suggestion. We agree that highlighting the broader international context particularly the substantial body of work in the USA will improve the paper's visibility and relevance for an international audience. In the revised manuscript, we have expanded the introduction to explicitly acknowledge parallel developments in the USA (including New England) and we have added the suggested references (and additional closely related works where appropriate) to better position our contribution within the global literature (Der Vaart et al., 2022; Suh et al., 2023; Bonhage et al., 2023).

2) Randomly, some references cited in the text are missing, e.g. Schneider et al. (2019, 2022) and Buras et al. (2015) in line 90 ff. Please check the entire text to ensure a correct and full list of references.

**Response**: Thank you for catching this. We will add the missing references and ensure that all in-text citations are fully matched with entries in the reference list ((Schneider et al., 2022; Schneider et al., 2019; Buras et al., 2020).

3) Line 112 ff: Perhaps this phrasing is slightly misleading because the 'preservation' is not only an effect of modern agriculture; RCHs were originally often situated in marginal landscapes with less productive soils, favouring forest use over crop production.

**Response**: Thank you for this helpful suggestion. We agree and have revised the text accordingly to clarify that the observed preservation pattern reflects not only the effects of modern agriculture, but also the original placement of many RCHs in marginal, less productive landscapes where forest use was favored over crop production:

„Previous studies have shown that hearths were rather preserved in less fertile soils, unsuitable for agriculture (Mikulski, 1994), than in areas with active cultivation. **Importantly, this pattern reflects not only differential preservation under modern land use, but also the original placement of many RCHs in marginal landscapes with less productive soils, where forest use was favored over crop production.** Due to that, many hearths were most likely destroyed by continuous farming practices like deep ploughing. Evidence for hearths in agricultural lands exists mainly in specific areas, such as in the Netherlands (Hardy et al., 2017)."

4) Line 189 ff: Please specify 'visually recognised'! Was identification carried out by different people, with manual picking and double-checking? Have you experienced any deviation between operators, especially with regard to false positives?

**Response**: In our workflow, "visually recognised" refers to manual identification on LiDAR-derived bare-earth DTM visualisations (hillshade, slope and local-relief/TPI). The mapping was carried out by one trained expert (Krzysztof Szewczyk) to ensure consistency. To check reliability and potential operator deviation, we conducted a cross-validation test in which five additional researchers independently interpreted selected grid cells using the same criteria; the results were fully consistent, indicating that the morphological signature and ruleset are robust under our guidelines. To minimise false positives, we applied strict, archaeology-informed criteria: an RCH was accepted only when it showed a round, flat-topped mound with a subtle peripheral rim, typically 6–25 m in diameter, and a minimum local relief of ~0.15 m (peak-to-shoulder) in the bare-earth DTM. Features that did not meet the mandatory shape/flat-top criteria were labelled ambiguous and excluded. We also screened for common confounders (e.g., forest clearings/canopy gaps) by relying strictly on bare-earth products and, where needed, checking orthophotos for signs of recent earthworks. While full field verification of all features is not feasible at this scale, during multiple field campaigns we checked accessible locations and, so far, all visited features classified as RCHs have been confirmed in the field.

5) Figure 2 and related data set: Why did you choose 25 km² and not 1 km²? There is already a wide variety of units used in the literature. To establish a consistent dataset on a European scale, a density unit of RCH per 1 sq m would be easier to handle and could be processed more easily in future projects. It would be good to provide the density data accordingly. Perhaps it is somewhere in the supplementary material that the reader cannot see.

**Response**: Thank you for this helpful suggestion. We chose 25 km² (5 × 5 km) for Figure 2 because the map is meant for national-scale visualization: a 1 × 1 km grid produces very high local variance due to strong clustering and results in a visually noisy pattern that is difficult to interpret on a country-wide figure. The 5 × 5 km resolution is therefore a compromise large enough to reduce pixel-level noise, but still fine enough to preserve regional contrasts. Importantly, the 500 × 500 m grid mentioned in the Methods was used only to ensure exhaustive manual interpretation and wall-to-wall coverage within forest polygons (CORINE Land Cover 2018), not for density estimation. To improve comparability and support future European-scale harmonisation, we have now also produced density layers aggregated to 1 × 1 km (1 km²) and we provide them in the Supplementary Material (below).

6) Line 273 ff: It is an interesting interpretation that the depressions should have been used for tar collection. Do you have physical proof from the RCH excavations, i.e. remains of tar in these depressions or on the RCH platforms? This interpretation could indeed explain the differences between RCH types with circular ditches and those with separated pits around a charcoal hearth. However, producing tar and charcoal simultaneously seems an unusual operation, and tar production sites have different architecture. The Fokt (2012) paper cited is not in the reference list. Therefore, it is not possible to investigate this further.

Response: Thank you for this comment. We agree that the tar-collection interpretation needs to be framed cautiously and supported with appropriate references. We will add the missing Fokt (2012) entry to the reference list and ensure full consistency between in-text citations and the bibliography. Regarding evidence, we do not claim that tar production can be demonstrated for the full national inventory. However, our targeted field investigations provide support for this hypothesis in selected locations: in several cases the morphology and sedimentary context of the depressions are consistent with the proposed function, as documented in our point-based fieldwork reported in Jonczak et al. (2024). We will explicitly cite this work and revise the text to clarify that the tar-related explanation is a working hypothesis supported by local observations, not a universal interpretation for all RCH types.

Once again, we thank the Reviewer for the supportive assessment and the thoughtful, actionable recommendations. We believe these changes significantly improve the manuscript's clarity, international framing, and methodological transparency, and we hope the revised version addresses all concerns satisfactorily.

References:

1. Bonhage, A., Raab, T., Schneider, A., Raab, A., Ouimet, W., Völkel, J., and Ramezany, S.: From site to state – Quantifying multi-scale legacy effects of historic landforms from charcoal production on soils in Connecticut, USA, Catena, 232, 10.1016/j.catena.2023.107426, 2023.
2. Buras, A., Hirsch, F., Schneider, A., Scharnweber, T., van der Maaten, E., Cruz-Garcia, R., Raab, T., and Wilmking, M.: Reduced above-ground growth and wood density but increased wood chemical concentrations of Scots pine on relict charcoal hearths, Sci Total Environ, 717, 137189, 10.1016/j.scitotenv.2020.137189, 2020.
3. der Vaart, W. V. v., Bonhage, A., Schneider, A., Ouimet, W., and Raab, T.: Automated large-scale mapping and analysis of relict charcoal hearths in Connecticut (USA) using a Deep Learning YOLOv4 framework, Archaeological Prospection, 30, 251-266, 10.1002/arp.1889, 2022.
4. Fokt, K.: Późnośredniowieczne osadnictwo wiejskie na Dolnym Śląsku w świetle badań archeologicznych, Księgarnia Akademicka, Kraków2012.
5. Jonczak, J., Barbarino, V., Chojnacka, A., Kruczkowska, B., Szewczyk, K., Gmińska-Nowak, B., Kołaczkowska, E., Łuców, D., Halaś, A., Mroczkowska, A., Słowińska, S., Kramkowski, M., Kowalska, A., and Słowiński, M.: Historical charcoal production as a factor in soil cover heterogeneity in a fluvioglacial landscape – A case study from northern Poland, Geoderma, 445, 10.1016/j.geoderma.2024.116892, 2024.
6. Schneider, A., Hirsch, F., Raab, A., and Raab, T.: Temperature Regime of a Charcoal-Enriched Land Use Legacy Soil, Soil Science Society of America Journal, 83, 565-574, 10.2136/sssaj2018.12.0483, 2019.
7. Schneider, A., Bonhage, A., Hirsch, F., Raab, A., and Raab, T.: Hot spots and hot zones of soil organic matter in forests as a legacy of historical charcoal production, Forest Ecology and Management, 504, 10.1016/j.foreco.2021.119846, 2022.
8. Suh, J. W., Ouimet, W. B., and Dow, S.: Reconstructing and identifying historic land use in northeastern United States using anthropogenic landforms and deep learning, Applied Geography, 161, 103121, https://doi.org/10.1016/j.apgeog.2023.103121, 2023.

Dear Reviewer 2,

Thank you very much for your thorough reading of our manuscript and for your positive evaluation of both the study and the open-access ReCHAR database. We sincerely appreciate your recognition of the novelty of a national-scale inventory and of the value of making the dataset public and expandable. We are also grateful for your constructive comments, in particular those concerning chronology and the minor technical corrections. Below, we respond to each point in detail and describe the changes made in the revised version.

**Specific comments:**

A key limitation, however, concerns the absence of chronological considerations as variable in the spatial analysis. All identified RCHs are implicitly treated as contemporaneous, although existing research (e.g., Rutkiewicz et al., 2021) demonstrates that this is unlikely. This assumption affects the reliability of the spatial and typological analyses that the paper aims to develop. Regional differences and variability in charcoal production practices cannot be analysed on morphological and environmental variables solely: the results risk being misleading without chronological differentiation.

In the conclusions, the authors themselves mention plans to incorporate additional data sources (e.g., toponymy, structures related with charcoal production), and I strongly encourage integrating a diachronic dimension (at least in selected sample areas). Stratigraphic investigations of RCHs, anthracological analyses, radiocarbon dating, and studies of the history of sampled forest environments would significantly strengthen interpretations and provide more reliable insights into the long-term dynamics of charcoal production and its environmental impacts.

Despite this limitation, the study offers significant potential and, in my view, is suitable for publication after minor revisions (few technical correction). Specifically, I recommend the following:

**Response**: Thank you very much for this thoughtful and constructive comment. On behalf of the authors, we fully agree that charcoal production in Poland spanned multiple centuries and that the mapped relict charcoal hearths (RCHs) cannot be assumed to be contemporaneous. We would like to clarify that we did not intend to imply synchrony; rather, the current dataset is a morphological and spatial inventory derived from LiDAR-based mapping, and for the vast majority of features we simply do not have direct chronological control. We therefore agree that our spatial and typological patterns should be interpreted as patterns of distribution and morphology, not as evidence for synchronous regional differences in production practices. At the same time, we agree with your suggestion to incorporate diachronic information where feasible. While it is not possible to estimate the age of all mapped sites, chronology can be established for selected, intensively studied locations. We will strengthen the Discussion/Outlook by explicitly outlining this pathway and by referring to existing dated and stratigraphically investigated examples, including the work by Rutkiewicz et al. (2021) and our own targeted field studies (e.g., Jonczak et al. (2024)). We will also emphasize that future work will focus on integrating stratigraphy, anthracology, radiocarbon dating, and forest/environmental history in sample areas, and then linking these dated case studies back to the national inventory.

**Specific comments:**

1) Bibliography

- Several references cited in the text appear missing from the bibliography (e.g., Buras et al., 2015; Miechówka and Drewnik, 2018; Fokt, 2012).

**Response**: Thank you for noticing this. We will add the missing references and ensure consistency between the in-text citations and the bibliography.

- Some inconsistencies in formatting should be corrected (e.g., capitalisation issues in lines 557 and 635; missing spaces between the date and the publisher/location, as in line 498).

**Response**: Thank you for noting these inconsistencies. We agree and will correct them in the revised manuscript, ensuring consistent formatting throughout.

2) Line 82

- "Massive" should be capitalised after the period.

Response: Thank you. This has been corrected.

3) Lines 167–168

- Remove the parentheses after "Lasota et al., 2018".

- Remove the repeated "et al., 2018" before "Miechówka and Drewnik, 2018)".

Response: Thank you for pointing out these typographical issues. We have corrected the citation formatting and removed the duplication.

**Once again, we thank the Reviewer for the encouraging assessment and for the constructive suggestions.**

References:
1. Jonczak, J., Barbarino, V., Chojnacka, A., Kruczkowska, B., Szewczyk, K., Gmińska-Nowak, B., Kołaczkowska, E., Łuców, D., Halaś, A., Mroczkowska, A., Słowińska, S., Kramkowski, M., Kowalska, A., and Słowiński, M.: Historical charcoal production as a factor in soil cover heterogeneity in a fluvioglacial landscape – A case study from northern Poland, Geoderma, 445, 10.1016/j.geoderma.2024.116892, 2024.
2. Rutkiewicz, P., Kalicki, T., Kłusakiewicz, E., and Przepióra, P.: Historical Charcoal Burning in the Kamienna River Basin (Old Polish Industrial District, Central Poland) – First Results from Lidar Data, Acta Geobalcanica, 8, 103-108, 10.18509/AGB218-3103r, 2021.

---

## Author Response (AR2)

Institute of Geography
and Spatial Organisation
Polish Academy of Sciences

Michał Słowiński, Prof. dr hab.
Department of Past Landscape Dynamics,
**Institute of Geography and Spatial Organization,**
**Polish Academy of Sciences**
January 9, 2026, Warszawa, Poland

*Dear Editor Robert Jackisch, Earth System Science Data (ESSD)*

Thank you very much for handling our manuscript and for guiding us through the review process. We also appreciate your positive assessment and the additional technical suggestions. Below, we respond to each point and indicate the corresponding changes made in the revised manuscript.

We have implemented all technical corrections suggested by the Editor. In addition, regarding the authorship change, I confirm that I have received written email confirmations from all co-authors agreeing to add Dr Mateusz Kramkowski as a co-author. As the corresponding author, I hereby confirm this on behalf of all co-authors.

Finally, thank you also for your private note. We appreciate the comment regarding the introduction; we are glad that, despite its length, it provides useful geographical context for a data paper.

With kind regards,
Michał Słowiński
on behalf of all co-authors